# Coulomb interactions between dipolar quantum fluctuations in van der Waals bound molecules and materials

Martin Stöhr [1], Mainak Sadhukhan[1,2], Yasmine S. Al-Hamdani[1,3], Jan Hermann [1] & Alexandre Tkatchenko [1]✉

Mutual Coulomb interactions between electrons lead to a plethora of interesting physical and chemical effects, especially if those interactions involve many fluctuating electrons over large spatial scales. Here, we identify and study in detail the Coulomb interaction between dipolar quantum fluctuations in the context of van der Waals complexes and materials. Up to now, the interaction arising from the modification of the electron density due to quantum van der Waals interactions was considered to be vanishingly small. We demonstrate that in supramolecular systems and for molecules embedded in nanostructures, such contributions can amount to up to 6 kJ/mol and can even lead to qualitative changes in the long-range van der Waals interaction. Taking into account these broad implications, we advocate for the systematic assessment of so-called Dipole-Correlated Coulomb Singles in large molecular systems and discuss their relevance for explaining several recent puzzling experimental observations of collective behavior in nanostructured materials.

[1] Department of Physics and Materials Science, University of Luxembourg, Luxembourg L-1511, Luxembourg. [2] Department of Chemistry, Indian Institute of Technology Kanpur, Kalyanpur, Kanpur 208 016, India. [3] Department of Chemistry, University of Zürich, CH-8057 Zürich, Switzerland. ✉email: alexandre.tkatchenko@uni.lu

Recent years have witnessed an ever-growing interest in nanostructured materials for sensor and filter applications, catalysis, or as energy materials[1–4]. This interest is nurtured by the manifold and highly tunable physicochemical properties of and within materials such as zeolites, hybrid organic-inorganic materials, metal or covalent organic frameworks, and layered materials[4–7]. A common feature among the above applications is that molecules tread into nanoscale voids and interact within confined spaces. The same applies to biomolecular systems, where processes often occur under the confinement of membranes or ion channels and interfacial water mediates the interaction among macromolecules[8–10]. The advancement of nanotechnologies and materials as well as our understanding of various biomolecular processes, thus, require a deep understanding of intermolecular interactions under (nano-)confinement.

Long-range van der Waals (vdW) dispersion, though typically characterized as "weak", represents a crucial part of these interactions and underpins much of the physical and chemical behavior in biology, chemistry, and materials science. VdW dispersion forces are a major part of the (dynamic) electron correlation energy and arise from quantum-mechanical fluctuations in the electronic charge distribution interacting via the Coulomb potential. Clearly, such Coulomb-coupled fluctuations are many-body in nature. Accounting for the many-body character of vdW interactions has been shown to play a decisive role in the accurate description of molecules and materials, in particular, with increasing system size and complexity[11–18]. Many-body treatments of dispersion forces in practically relevant systems are typically carried out within the interatomic dipole limit or the random phase approximation (RPA). Effects beyond these methods are rarely investigated and usually ad hoc considered to be negligible.

Lately, this notion is increasingly disputed, however, as the missing physics in the most widely used computational methods continue to stand in the way of understanding a growing number of state-of-the-art experimental phenomena. For example, the experiments of Pollice et al.[19] reveal a considerable impact of the solvent on the intra- and intermolecular dispersion interactions in proton-bound dimers, whereas their computational study using implicit solvents in combination with methods based on atom-pairwise dipolar vdW interactions fails to capture the effect. In another related experiment, Secchi et al.[20] found that water flows ultra-fast through narrow carbon nanotubes (CNTs), but not through boron nitride nanotubes. In this regard, theorists are still working toward satisfactory modeling of this effect and capturing the underlying physical interactions in vdW materials[21–24]. In a similar vein, the spatial separation and ordering of large polarizable molecules on metal surfaces[25,26], salient in organic thin films for organic electronics, highlights gaps in common modeling approaches. For instance, Wagner et al.[25] showed that large aromatic molecules organize into highly ordered arrays at high coverage on Au(111) and interestingly, the authors rationalize their findings using repulsive, Coulombic intermolecular interactions, induced by electronic screening from the metal. Such puzzling experimental observations and the various phenomena emerging under nanoscale confinement[27–31] challenge our current understanding of intermolecular interactions in complex systems and suggests to reconsider the contribution from vdW forces beyond the common dipole approximation or RPA.

In previous work, we have introduced a formalism to account for beyond-dipolar vdW interactions and exemplified the groundbreaking effect it can have for two oscillators with reduced dimensionality, representing a model system for confined atoms or molecules[32]. Our work here represents the applicable extension of this formalism to atomistic modeling and provides a consistent and practical approach to incorporate higher-order terms of vdW forces while retaining a full many-body treatment based on the many-body dispersion (MBD) framework[33]. We highlight the important role of this contribution, here referred to as dipole-correlated Coulomb singles (DCS), and discuss its complex quantum-mechanical character. Finally, beyond-dipolar many-body treatment of vdW interactions allows us to show the non-trivial behavior of vdW interactions inside nanoscale structures and to elaborate on the physical interactions indicated by the above-mentioned experiments[19,20,25,26].

## Results

VdW dispersion interactions originate from the long-range (dynamic) electron correlation energy. Therefore, they depend solely on the fluctuations in the instantaneous electronic charge distribution, and more specifically, on correlations in those fluctuations that occur on length scales exceeding intra-atomic distances. Taking advantage of this, the MBD formalism models vdW interactions by approximating the charge fluctuations of a given electronic system with a set of effective quantum harmonic (Drude) oscillators; each oscillator corresponding to a single atom, mimicking its long-range electrodynamic response, and mutually interacting with other oscillators via the dipole potential, $T_{pp}$. Such a coupled oscillator model represents an efficient and reliable approach for describing electronic polarizabilities[34–37] exactly reproducing the leading-order response of real atoms[38] and accurately capturing polarization effects as well as vdW dispersion in molecules and materials[39–41]. In addition, it can be used to describe excess electrons in matter[42] and has been shown to reproduce dispersion-polarized electron densities[16] and the relation between the electronic polarizability and geometry in vdW bound dimers as obtained for real atoms[43]. The Hamiltonian for a set of dipole-coupled oscillators can be written as

$$\hat{\mathcal{H}}_{DC} = \hat{\mathcal{T}} + \hat{U} + \hat{T}_{pp} \equiv \hat{\mathcal{H}}_0 + \hat{T}_{pp} \qquad (1)$$

where $\hat{\mathcal{T}}$ and $\hat{U}$ are the kinetic energy and harmonic potential operators, respectively. This Hamiltonian lends itself to a closed-form solution via eigenmode transformation and the vdW energy is obtained as the difference in the ground-state energy between the dipole-coupled system and its non-interacting variant (described by $\hat{\mathcal{H}}_0$),

$$E_{MBD} = E_{DC} - E_0 = \sum_k \frac{\omega_{DC,k}}{2} - \sum_k \frac{\omega_{0,k}}{2} \qquad (2)$$

where $\omega_k$ is the effective frequency of the $k$th oscillator mode in the corresponding system[33,34]. When using an oscillator at every point in space, the dipole-coupled framework can essentially describe any response allowed by quantum field theory, but in a coarse-grained formalism of atomic response (like the MBD approach), one needs to account for beyond-dipolar couplings.

**Methodology**. In this work, we address this issue by going beyond the dipole approximation in $\hat{\mathcal{H}}_{DC}$ while retaining an efficient atomistic formalism. Given that the equivalent of Equation (1) with the full Coulomb interaction does not allow for a straightforward closed-form solution, we derive and evaluate the correction toward full Coulomb coupling, $\hat{V}' = f_{damp}(\hat{V}_{Coul} - \hat{T}_{pp})$, to first order in perturbation theory,

$$E_{DCS} = \langle \Psi_{DC} | \hat{V}' | \Psi_{DC} \rangle - \langle \Psi_0 | \hat{V}' | \Psi_0 \rangle, \qquad (3)$$

where $|\Psi_{DC}\rangle$ and $|\Psi_0\rangle$ represent the ground-state of the dipole-coupled and non-interacting system as described by $\hat{\mathcal{H}}_{DC}$ and $\hat{\mathcal{H}}_0$, respectively. The second term in Equation (3) describes the non-zero mean-field part of the beyond-dipolar interaction between

Drude oscillators[44] and is necessary to retain pure correlation and correlation-induced contributions in $E_{DCS}$[45]. In the spirit of the terminology of quantum-chemical expansion series such as coupled-cluster theory, we will refer to the first-order full Coulomb correction over the MBD energy as DCS. Given that the zeroth order (MBD) Hamiltonian already represents a correlated state within dipole coupling, the present singles term is to be distinguished from those in post-Hartree-Fock methods or the RPA, where the zeroth-order theory corresponds to a mean-field, uncorrelated state. A more detailed discussion on DCS in the context of correlated electronic-structure methods is given in the final section of this work.

Typically, the electronic Coulomb energy is divided into its classical and correlation parts in electronic-structure theory. Likewise, we also divide the oscillator Coulomb energy into its classical component, $J[\rho]$, and the correlation energy, $E_{corr}[\Psi]$, and denote $E_{dip}[\Psi] = \langle\Psi|\hat{T}_{pp}|\Psi\rangle$. As $E_{corr}[\Psi_0] = E_{dip}[\Psi_0] = 0$, the first-order full Coulomb contribution can be written as

$$E_{DCS} = \left(J[\rho_{DC}] - J[\rho_0]\right) + \left(E_{corr}[\Psi_{DC}] - E_{dip}[\Psi_{DC}]\right). \quad (4)$$

Here, the first term on the right is the change in the electrostatic energy of the oscillators caused by the polarization of the system, which is itself induced by vdW dispersion interactions. The second term on the right is a Coulomb correction to the dipole correlation energy. To first order in $\Delta\rho = \rho_{DC} - \rho_0$, the first term can be expressed as $J[\rho_0, \Delta\rho]$, i.e., the electrostatic interaction energy of the non-interacting oscillator densities with the density polarization induced by the dispersion. $E_{DCS}$ can therefore also be seen as a dispersion–polarization coupling energy plus beyond-dipolar interaction between quantum fluctuations. The beyond-dipolar contribution is thereby evaluated on the density of dipolar quantum fluctuations and thus in contrast to conventional higher-order multipolar interatomic vdW interactions. In Fig. 1, we provide a schematic representation of the DCS interaction.

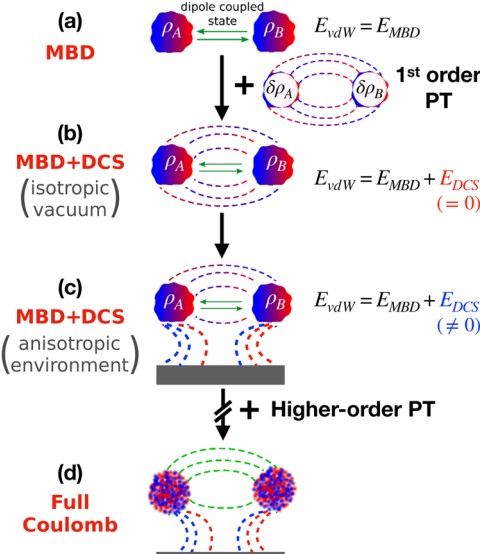

**Fig. 1 Schematic representation of dipole-correlated Coulomb singles (DCS). a** Green arrows represent dipole coupling between electronic fragments. First-order perturbation theory (PT) on top of the many-body dispersion formalism (MBD) captures the interaction energy, $E_{DCS}$, between $\delta\rho_A$ and $\delta\rho_B$, depicted by field lines. **b** $E_{DCS}$ vanishes in 3D isotropic vacuum because of symmetry. **c** Under rotational symmetry-breaking confinement, electric field lines between electronic fragments deform, which leads to $E_{DCS} \neq 0$. **d** Further inclusion of higher-order terms leads to full Coulomb-coupled vdW interaction.

As can be seen from this schematic illustration, the presented formalism reaches the well-defined limit of fully Coulomb-coupled oscillators with increasing orders of perturbation theory. Thus, DCS contributions do not represent an ad hoc correction of dipolar many-body dispersion, but the leading term toward the well-established and reliable quantum Drude oscillator model of vdW dispersion[39,40], while requiring only a fraction of its computational costs.

We note that in the presence of a polarizable environment, the DCS interaction between two bodies can decay asymptotically slower than the zeroth-order MBD energy. For example, the dipole correlation in the MBD ground-state wavefunction between a finite body and its environment induces permanent quadrupole moments on the oscillators, causing the resulting interaction to decay as $R^{-5}$. In contrast, the MBD interaction energy between two bodies in such a system decays asymptotically as $R^{-6}$. The physical origin of $R^{-5}$-dependent repulsive interactions between oscillators with reduced dimensionality as reported in ref. [32] has been controversially debated[46,47]. The generalized explanations and results for realistic systems reported in this publication finally resolve this discussion and clearly show that this leading-order behavior is not owing to purely electrostatic interactions, but originates from dispersion-induced electron density polarization effects. From this discussion, it is also clear that the DCS contribution is a leading-order term, which in general could be comparable to or even more important than the renowned higher-order multipolar terms (dipole–quadrupole and higher terms[44]) in the vdW dispersion energy. Higher-order terms in the perturbation theory used here are naturally lower in magnitude and show a comparably quick decay with interatomic separations[32]. This justifies the limitation to the more-slowly decaying first-order DCS term.

The accurate prediction of interaction energies for non-covalently bound materials is an ongoing challenge for researchers and workhorse density-functional theory (DFT) methods are at the forefront of development efforts. The complex balance of intermolecular interactions is particularly challenging to predict and what is more, the target accuracy is generally in the range of a few kJ/mol in interaction energies. Our approach to better accuracy is to improve the physical basis of theoretical methods—here, by incorporating the DCS contributions. First, we combine DCS with MBD in MBD+DCS to compute interaction energies of small molecular dimers. Following this, MBD+DCS is used to compute the binding energies of supramolecular host–guest complexes and confined Xe dimers in capped CNTs. These larger confined systems reveal the impact of $E_{DCS}$ on binding energies, as well as the length scale and character of the emergent changes to long-range interactions.

**DCS in small molecular dimers**. We begin by applying the first-order perturbation term (3) to the S66 data set[48] of small, unconfined molecular dimers. For such systems, semi-local or hybrid density-functional approximations in conjunction with the MBD formalism are designed to provide excellent agreement with accurate reference results from coupled-cluster calculations with single, double, and perturbative triple excitations (CCSD(T))[33]. The S66 set contains non-covalently interacting dimers in 3D isotropic vacuum and in accordance with Fig. 1, we expect minuscule first-order Coulomb corrections in this case. Indeed, we find that the DCS contributions for all systems in S66 are very small and they can have both positive as well as negative values (see Supplementary Fig. 2). As a result, the accuracy of vdW-inclusive DFT remains equally good upon account for DCS contributions to the interaction energies of small molecular dimers as contained in S66.

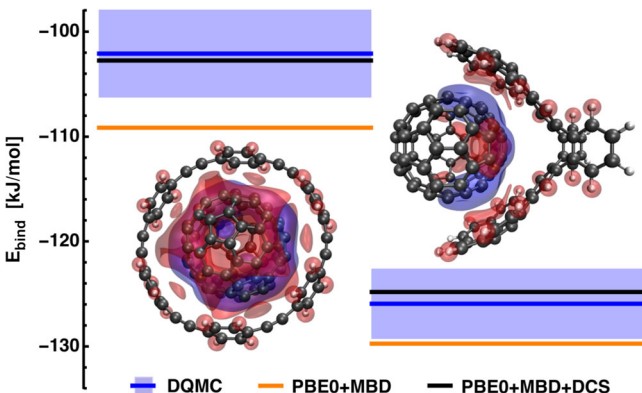

**Fig. 2 Binding energies and dispersion–polarization of a $C_{70}$ fullerene different host molecules.** Host molecules: 6-CPPA (left), "buckyball-catcher" (right). PBE0+MBD results (orange), diffusion quantum Monte-Carlo reference (DQMC, blue line; error bars shown as boxes), PBE0+MBD including dipole-correlated Coulomb singles (DCS, black line). DQMC reference data were taken from ref. [16]. The depiction of the complexes includes iso-surfaces at ±0.003 (a.u.) of the change in the density of electronic fluctuations with respect to the isolated monomers (red: decrease, blue: increase).

**Coulomb corrections for host–guest complexes.** Host–guest molecular systems are significantly more complex than the S66 dimers, but are still tractable with accurate benchmark methods such as diffusion quantum Monte-Carlo (DQMC). Here, we first ascertain the impact of DCS on the binding energies of host–guest complexes and demonstrate the accuracy of PBE0+MBD+DCS with respect to DQMC reference interaction energies from previous works. Fig. 2 shows two examples of host–guest complexes. For both systems, the guest molecule is a $C_{70}$-fullerene (buckyball) while the host is either [6]-cycloparaphenyleneacetylene (6-CPPA, Fig. 2 left) or the "buckyball-catcher" molecule (Fig. 2 right)[49]. In the framework of Fig. 1, the host molecule serves as both confinement and the interaction partner. The 6-CPPA and catcher molecules provide a different confining environment for the buckyball by virtue of their geometry. We therefore focus on these systems to showcase the contribution of $E_{DCS}$ in confined systems.

Within the dipole approximation, it has already been shown that many-body effects have an important role in the description of binding energies in such host–guest complexes[16]. Here, we show that also interactions beyond the dipolar MBD, in the form of DCS, have a significant effect and that inclusion of DCS to PBE0+MBD yields excellent agreement with the reference results from DQMC. The DCS contribution for $C_{70}$ in 6-CPPA and in the buckyball-catcher is 6.4 kJ/mol and 4.9 kJ/mol, respectively. Considering our findings for the S66 data set above, this clearly highlights the importance of DCS corrections once the 3D isotropy of vacuum is substantially perturbed.

The relative contribution of DCS to the total binding energy for $C_{70}$ in 6-CPPA and in the buckyball-catcher is 6.2% and 3.9%, respectively. As can be seen from Fig. 2 (and more clearly from Fig. 3 below), the DCS contribution does not correlate with the system size nor the vdW interaction within the dipolar approximation. However, the different $E_{DCS}$ contributions can be interpreted in terms of the physics described in the dipole-coupled state, which represents the starting point (unperturbed state) for the calculation of the DCS contribution. One important factor is how the dipolar coupling changes the density of electronic fluctuations (i.e., $\delta\rho$ shown in Fig. 1), which can be obtained as the expectation value of the charge density operator via the wavefunction of the dipole-coupled, $\Psi_{DC}$, and

non-interacting set, $\Psi_0$, of quantum harmonic oscillators. Analysis of the difference of $\delta\rho$ in the host–guest complex with respect to the isolated monomers gives us a measure of how much the density deforms upon the host–guest interaction and, therefore, forms a connection between confinement and electronic fluctuations and polarizability. The density difference shown in Fig. 2 shows that, upon dipole coupling, the density of electronic fluctuations for $C_{70}$ in 6-CPPA is more strongly deformed into the plane of 6-CPPA. Furthermore, we find that the overall displaced charge, i.e., the integral over the absolute density difference, can serve as a descriptor for the DCS contribution to the interaction energy: with increasing displaced charge, we observe an increased DCS contribution to the interaction energy. We point out that this also applies to the systems considered below. Hence, the electronic properties obtained for the dipole-coupled state can serve as a qualitative rule-of-thumb to estimate the magnitude of $E_{DCS}$.

**Asymmetry and steric effects.** To explore the connection between $E_{DCS}$ and confinement, we analyze a set of geometrically similar ring–$C_{70}$ complexes as depicted in Fig. 3: the four complexes are $C_{70}$ hosted by four different conformations of 8-CPPA. In previous work, PBE0+MBD has been shown to provide reasonably accurate binding energies with respect to DQMC[16]. As can be seen from Fig. 3A, the addition of the DCS contribution to PBE0+MBD further improves the binding energies of all four complexes. However, the individual DCS contributions vary significantly across these conformations (see relative $E_{DCS}$ as shown in Fig. 3B).

In order to study the potential role of asymmetry and steric effects for $E_{DCS}$, we define two geometrical measures: one for proximity ($f_d$), which is given by the sum of inverse distances between the atoms of the fullerene guest molecule and the CPPA-host, and one for the asymmetry of the system ($f_a$). For the latter, we define a plane along the elongated axis of $C_{70}$ that is perpendicular to the CPPA ring (labeled $P_v$ plane in Fig. 3C). The four phenyl units closest to this plane are considered axial and the remaining four radial. Based on this classification, we define axial vicinity ($A_\parallel$) and radial vicinity ($A_\perp$) by summing the inverse distances between all fullerene atoms and atoms of the axial and radial phenyl rings, respectively. Our measure of (axial–radial) asymmetry is then given by $f_a = (A_\parallel - A_\perp)/(A_\parallel + A_\perp)$. Fig. 3B summarizes the results for the proximity and asymmetry measures and the ratio of $E_{DCS}$ of each system and that of R4 ($f_e = E_{DCS}(R_i)/E_{DCS}(R_4), i = \{1, 2, 3, 4\}$). It is clear that in principle, proximity has a role in electronic confinement (cf. proximity and DCS contributions for $C_{70}$ in 6-CPPA and R1). As can be seen from the detailed analysis in Fig. 3B, however, purely geometric considerations do not correspond directly to the trends in $E_{DCS}$. First, the proximity measure, $f_d$, is insensitive to the different confining environments and remains almost constant among all four conformations, whereas the DCS contribution varies significantly. Furthermore, the asymmetry measure, $f_a$, has no correlation with $E_{DCS}$ (see R2 versus R3 and R4 in Fig. 3B). Thus, also a pairwise description of asymmetry between atomic positions is insufficient to predict the qualitative trend of the contribution of DCS.

The failure to capture the behavior of $E_{DCS}$ in terms of simple geometric characteristics stems from the fact that the DCS contribution is a quantum-mechanical effect arising from long-range electron correlation, which shows a non-trivial dependence on the geometrical features of a system. A considerable part of $E_{DCS}$ represents charge polarization effects due to long-range electron correlation, cf. Equation (4).

As discussed for the previous complexes, the displaced charge within MBD (as depicted in Fig. 2) can provide a measure for the

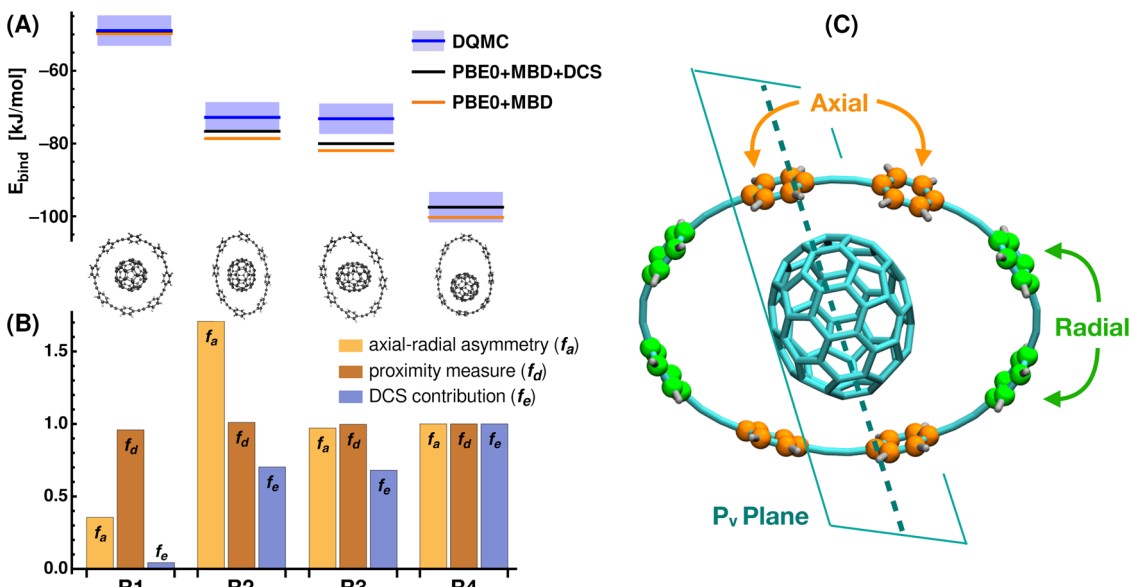

**Fig. 3 Dipole-correlated Coulomb singles (DCS) contributions to binding energies of ring–$C_{70}$ complexes and correlation to structural features.**
**A** Binding energies for four ring–$C_{70}$ host–guest complexes (R1–R4): PBE0+MBD results (orange), diffusion quantum Monte-Carlo reference (DQMC, blue line; error bars as boxes), PBE0+MBD including DCS (black line). DQMC reference data taken from ref. [16]. The hosts for R1–R4 are 8-CPPA rings.
**B** Measure of axial–radial asymmetry ($f_a$), proximity measure ($f_d$), and DCS contribution ($f_e$) to binding energy (all values normalized to results for R4). Definition of axial and radial phenyl units of 8-CPPA via $P_v$ plane shown in **C**.

dispersion–polarization-like term and indeed tracks the qualitative trend in the relative DCS interaction energies for all supramolecular complexes treated here. The best geometry-based metric found in this work is a sum of inverse distances to the power of five, which resembles an interaction of quadrupoles induced by long-range correlation (vide supra). Further information on qualitative descriptors can be found in Supplementary Fig. 3.

**DCS in nanotubes.** Having established the importance of $E_{DCS}$ for confined host–guest systems, we now employ our methodology to investigate the effects of confinement for a Xe dimer inside CNTs. In particular, we answer the question how the presence and strength of confinement changes the importance of the DCS correction relative to dipolar vdW interactions. Xe does not possess permanent multipoles and has substantial polarizability. As a result, the long-range Xe–Xe interaction has pure vdW character.

CNTs can be generally classified according to their chiral indices $(m, n)$, where armchair CNTs with $m = n$ are metallic in nature. As the DCS contribution becomes especially important in the presence of metallic screening[32], we analyze the Xe–Xe interaction inside two armchair, hydrogen-capped CNTs with $m = n = 5$ and $m = n = 6$. The diameters of the (5, 5)- and (6, 6)-CNTs are 6.78 Å and 8.13 Å, respectively. The length of each nanotube was chosen to be 30 Å, which is sufficient to avoid any significant edge effects. The binding energies of the Xe dimer inside the nanotubes are calculated as $E_{int} = E_{Xe_2(NT)} + E_{CNT} − E_{Xe_A(NT)} − E_{Xe_B(NT)}$, where $E_{Xe_2(NT)}$, $E_{Xe_{A/B}(NT)}$, and $E_{CNT}$ are the energies of $Xe_2$ inside the CNT, the two single Xe atoms hosted by the nanotube, and the bare CNT, respectively.

Fig. 4 summarizes the effect of the confining potential of capped CNTs on the Xe dimer binding energy. We focus on the variation of the dipole-coupled MBD and the corresponding DCS contributions as a function of the inter-Xe distance, $R$. In Fig. 4A, we show the effect of confinement on the individual contributions

by comparing a Xe dimer inside the (6, 6)-CNT and in gas-phase. One clearly sees that both $E_{MBD}$ as well as $E_{DCS}$ become less attractive owing to confinement. In the case of the MBD interaction energy this can be attributed to (i) decreased Xe polarizabilities due to the screening by the CNT and (ii) the restriction of electronic fluctuations on the Xe atoms due to correlation with fluctuations in the CNT. This reduction of the vdW interaction as a result of many-body correlation has been observed and detailed in a number of previous works[12,13,16,33]. It is notable that the bare presence of the confinement affects $E_{MBD}$ more strongly than the DCS interaction energy.

Fig. 4B then shows the MBD and DCS components for the two CNTs with different radii. In contrast to the bare presence of confinement, the type and strength of the confinement, as represented by the different nanotubes, has a larger effect on the DCS interaction than on the MBD contribution. We also note that the effect of the different environment on the MBD interaction is negligible after 6 Å and the binding curves follow the same behavior, whereas the effect is more long-ranged for $E_{DCS}$. DCS are more sensitive to the characteristics of the confinement compared to $E_{MBD}$. As expected, the destabilization of the Xe dimer owing to screening and many-body correlation effects is less pronounced inside the (6, 6)-CNT.

Fig. 4C shows the cumulative vdW binding energy $E_{vdW} = E_{DCS} + E_{MBD}$. In total, confinement in a CNT leads to a substantial decrease of the Xe–Xe vdW interaction. Mostly owing to the DCS contribution, this destabilization is strongly dependent on the confining environment. This shows that the total long-range vdW interaction can be in fact substantially altered by (nano-)confinement, whereas the bare MBD treatment would predict the environment to have no effect beyond interatomic distances of 6 Å. For $Xe_2$ inside a (5, 5)-CNT, the interplay of the repulsive DCS contribution and the attractive MBD interaction interestingly leads to a near-linear behavior for separations of 6 Å to ~8 Å. To explore the balance between the repulsive $E_{DCS}$ and the attractive $E_{MBD}$ more clearly, we show the absolute value of their ratio as a function of $R$ in the inset of Fig. 4C. In all cases, the ratio, i.e., the relative importance of the

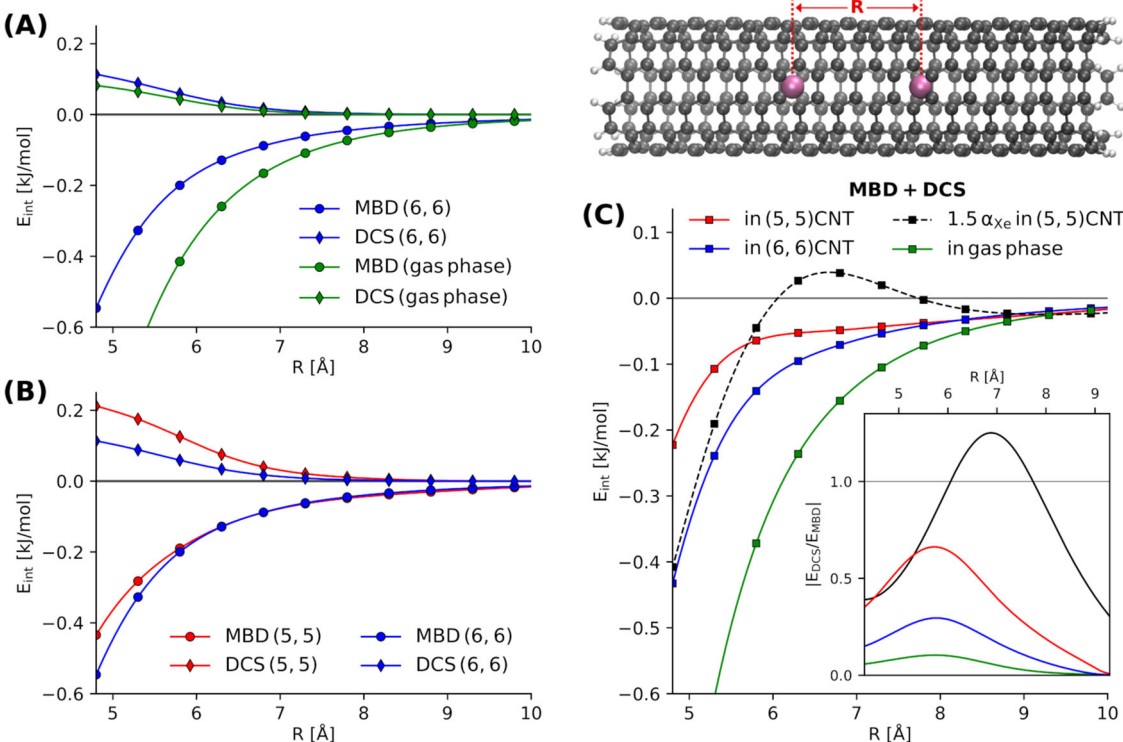

**Fig. 4 MBD and dipole-correlated Coulomb singles (DCS) contributions to the Xe–Xe interaction inside carbon nanotubes (CNTs). A** Comparing the MBD and DCS contributions inside a (6, 6)-CNT and in gas-phase as a function of the Xe–Xe separation, R. **B** Effect of the different confinements of a (5, 5)- and (6, 6)-CNT on $E_{MBD}$ and $E_{DCS}$. **C** Two Xe atoms (violet) encapsulated in a CNT. Total van der Waals interaction energy given as sum of $E_{MBD}$ and $E_{DCS}$ including the results when increasing the Xe polarizability by 50% (black). The inset shows the variation of the absolute value of the ratio of $E_{DCS}$ and $E_{MBD}$.

DCS contribution, increases with larger inter-Xe distances and reaches a maximum at a distance of ~5.8 Å before converging to zero. The interatomic distance at which we observe the maximum is surprisingly independent of the presence and strength of the confinement. The ratio of $E_{DCS}$ and $E_{MBD}$ and its maximum value, on the other side, strongly depends on the environment of the interacting particles.

In order to highlight the critical role played by the response properties of the objects interacting under confinement, we have performed the analysis for $Xe_2$ inside a (5, 5)-CNT while increasing the Xe polarizability by 50% (black curve). The results indicate a pivotal role of the polarizability in the total vdW interaction in confined systems: at shorter interatomic distances, the overall interaction is increased as the attractive MBD contribution is affected more strongly. In the very long-range limit, the interaction converges to the same behavior as for normal $Xe_2$. In the intermediate region, however, the repulsive contribution from DCS increases more strongly than its MBD counterpart, which leads to a substantial destabilization and eventually repulsive interaction energy. The interplay between $E_{DCS}$ and $E_{MBD}$ in this intermediate region gives rise to a maximum followed by a very shallow minimum creating a small barrier of ~0.1 kJ/mol in the binding curve. All this can be explained by a much higher sensitivity of $E_{DCS}$ to the Xe polarizability compared to the MBD interaction energy. Accordingly, changing the polarizability has a strong effect on the ratio of $E_{DCS}$ and $E_{MBD}$ and its maximum (see Fig. 4C). The ratio surpasses 1, meaning that $E_{DCS}$ supersedes the MBD contribution in magnitude and introduces a region of repulsive interaction at ~7 Å. The position of the maximum is thus increased by almost 1 Å compared to normal $Xe_2$. Altogether, we can conclude that

with increasing polarizability, the relative importance of the DCS contribution increases, becomes more long-ranged, and can lead to non-trivial qualitative changes in the overall vdW interaction. For the considered system of $Xe_2$ inside CNTs, the vdW interaction thereby fully governs the total long-range interaction. The corresponding PBE-DFT interaction energies are of negligible magnitude at inter-Xe distances beyond ~6 Å and only introduce the well-known repulsive contributions at shorter separations. As a result, the qualitative changes owing to DCS reported above remain unaltered by inclusion of contributions captured in semi-local DFT. One particularly interesting aspect of the total PBE+MBD+DCS interaction is that PBE contributions together with DCS cancel out the meta-stable state of $Xe_2$ in the (5,5)-CNT. On the level of PBE+MBD, one observes a local minimum at a inter-Xe distance of ~5.5 Å, whereas PBE+MBD+DCS predicts only repulsive or negligible interaction at all distances. The corresponding PBE-DFT and total interaction energies are summarized in Supplementary Fig. 4.

## Discussion

In this work, we introduce DCS as a distinct component of the interaction energy whose description is missing in standard vdW-inclusive DFT, for which we develop an explicit model within the MBD framework, and demonstrate that it can have a significant effect on vdW interactions in supramolecular systems and under nano-confinement. There are three main reasons why $E_{DCS}$ has not been addressed before. First, the resolution and accuracy of experimental setups have not been sufficient to reveal the unusual behavior arising from $E_{DCS}$ at the microscopic length scale. For example, the nano-fluidic techniques and manufacturing of

nanotubes with desired properties, which helped reveal the phenomenon of accelerated water flow through carbon nanotubes, have become available only recently. Second, the prevalent conception about the universality of long-range attraction between polarizable moieties has subdued explanations of the observed experimental phenomena that would accommodate long-range repulsive forces. Third, although ab initio electronic-structure methods such as coupled-cluster theory or quantum Monte-Carlo inherently describe DCS, the prohibitively high computational cost of such methods for larger systems did not allow for the fine analysis as enabled by our efficient approach. The computational costs of the presented DCS formalism without approximations scale with the fifth power of the number of atoms. However, this is accompanied by a very small prefactor and as a result, the computation of DCS produces negligible additional costs to semi-local or hybrid DFT calculations for systems of up to several hundred atoms. The present formalism further solely relies on the MBD wavefunction, which in turn is based on the definition of atomic polarizabilities within a molecule or material. So, the DCS formalism could equally well be included in force field calculations as presented previously for the MBD model[50]. Although the remaining computational costs limit its application in molecular dynamics simulations, DCS can be used to improve the description of structural ensembles via energy reweighting. In addition to such a posteriori corrections, our DCS formalism enables the determination of improved effective interatomic potentials for complex systems.

In order to fully capture $E_{DCS}$, an electronic-structure method has to describe its classical, dispersion–polarization-like term and correlation beyond interatomic dipole-dipole interactions (see Equation (4)). In the language of coupled-cluster theory, the former requires a fully self-consistent coupling between singles and doubles. As such, CCSD and beyond do capture both of these components. Only treating doubles amplitudes as in CCD or purely perturbative treatment of doubles does not. Quantum Monte-Carlo in principle provides a full solution of the many-electron Schrödinger equation and thus fully includes DCS. Also, the quantum Drude oscillator model with full Coulomb coupling[39,40,51] captures $E_{DCS}$. Symmetry-adapted perturbation theory includes the correlation component of DCS for intermolecular interaction, but dispersion–polarization contributions only appear beyond the typical limitation to second order. Given that the charge density polarization induced by long-range correlation leads to a slower decay with the distance to nuclei[16,52], all of the above require sufficiently large basis sets, which further increases their already high computational costs.

From the approximate electronic-structure methods applicable to larger systems, ordinary RPA, as commonly used to study layered materials, captures the full Coulomb interaction, but neglects the singles-like effect of the long-range electron correlation on the one-electron orbitals. This can further be seen from the equivalency of RPA and CCD within ring-diagram approximation[53]. One promising route toward capturing DCS is a fully self-consistent treatment of electron correlation within the RPA. However, current implementations of this approach, such as the self-consistent $GW$ method[54], do not yet provide an accurate description of dispersion-induced electron density polarization[52] and thus require further developments. In second-order Møller-Plesset perturbation theory (MP2), the effect of long-range correlation on the wavefunction is not reflected in the energy and thus MP2 does not cover $E_{DCS}$. Evaluating singles(-like) contributions on top of a long-range correlated wavefunction as obtained from MP2, however, does allow to recover $E_{DCS}$. Accounting for single excitation contributions within RPA as presented by Ren and co-workers[55], on the other side, does not as

it is based on a (mean-field) DFT wavefunction. Conventional (semi-)local density-functional approximations neglect long-range vdW interactions entirely and the standard DFT+vdW approaches, including those that go beyond interatomic dipole-dipole interactions, such as Grimme's D3[56] or the exchange-hole dipole moment (XDM) model[57–60], do not account for $E_{DCS}$. Incorporating vdW functionals into DFT in a self-consistent fashion[52] may recover the electron density component of $E_{DCS}$, but will not capture the full Coulomb component, complementary to ordinary RPA. The limited correlation with geometric descriptors (see Fig. 3) and complex changes to the interaction of $Xe_2$ inside CNTs finally show that the so-far neglected effect of DCS can neither be described phenomenologically via trivial modifications to the damping function or polarizability model, but need to be accounted for on a physical, methodological level.

In summary, we developed a consistent, unified methodology to incorporate a previously neglected part of the full Coulomb coupling between instantaneous electronic fluctuations within a quantum-mechanical many-body treatment of vdW interactions. We show that the inclusion of this contribution becomes significant for relatively larger molecular systems and can even change the qualitative nature of intermolecular interactions. The negligible computational cost of the present methodology compared with benchmark electronic-structure methods allows us to explore the emergent role of beyond-dipolar, beyond-pairwise vdW interactions in large-scale systems. Our surprising results for the interaction of a Xenon dimer inside carbon nanotubes (Fig. 4) suggests a possible explanation for the high flow rate of water through nanotubes, by way of modulation of the water polarizability through short-range intermolecular interactions[61], which in turn reduce the long-range vdW interaction. Careful study of the mutual interplay of such effects and further well-defined reference data from methods that incorporate DCS may be necessary to fully explain such puzzling effects at the nanoscale and we consider the present work to be the first step in this direction.

## Methods

In accordance with Equation (3), the DCS contribution to the vdW energy can be calculated from the beyond-dipole potential and the wavefunctions of dipole-coupled and uncoupled quantum (Drude) oscillators, respectively. These wavefunctions can be obtained directly from solving the MBD Hamiltonian (1) (for further details, see ref. [16]). With full Coulomb coupling the vdW dispersion energy is well-behaved in all cases. Using only dipolar or beyond-dipolar coupling individually, however, leads to a divergence of the vdW energy at short distances. The perturbing potential in the proposed formalism, $V'$, is therefore given by the damped beyond-dipolar potential, i.e., the sum over $f_{damp}^{AB} (V_{AB}^{Coul} - V_{AB}^{dip})$ for all pairs of oscillators $A$ and $B$ with $f_{damp}^{AB}$ as a damping function. The damping function for $E_{DCS}$ thereby follows the same Fermi-like functional form as in MBD[33], where the parameters $a$ and $\beta$ have been set to 10.12 and 1.4, respectively. This choice of the parameters provided robust results for all systems studied in this work. Optimal tuning of the damping function, however, requires an increased availability of accurate reference data for larger-scale systems such as the confined Xe dimer studied here, where $E_{DCS}$ has a significant role. As such, a more thorough investigation and optimization of the damping function is subject to ongoing work. Calculation of $E_{DCS}$ was carried out within the libMBD software package[62].

All DFT calculations presented in this work have been carried out within the FHI-aims package[63]. For the calculations employing the PBE0 hybrid functional, results at the default really tight level of settings have been extrapolated based on PBE0 with tight settings and results obtained with the PBE functional with tight and really tight settings. Although significantly reducing computational costs, this scheme has been proven to provide an excellent estimate for PBE0 at the really tight level[17]. The same extrapolation scheme was used to account for the effect of the corresponding change in Hirshfeld volumes, which form the basis for all MBD and DCS calculations. All results reported on Xe inside CNTs were obtained using the PBE functional with the tight level of settings in FHI-aims.

## Data availability

The data presented in this publication are available from the authors.

## Code availability

The libMBD package[62] used to calculate the contributions of Dipole-Correlated Coulomb Singles is available at https://github.com/jhrmnn/libmbd.

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

## Acknowledgements

The authors acknowledge many helpful discussions with Igor Poltavskyi, Max Schwilk and Johannes Hoja. M. Stöhr thanks the Fonds National de la Recherche Luxembourg (FNR) for financial support under AFR PhD Grant 11274975 (CNDTEC). Y.S.A. acknowledges funding provided by NIH grant number R01GM118697. A.T. and M. Sadhukhan were supported by the European Research Council (ERC-CoG BeStMo) and A.T. further acknowledges financial support from the FNR-CORE program C16/MS/11360857 (QUANTION). The results presented in this work have been obtained using the HPC facilities of the University of Luxembourg.

## Author contributions
The work was designed and conceived by A.T. with contributions from Y.S.A., M. Sadhukhan and M. Stöhr. M. Sadhukhan and J.H. formulated and implemented the presented methodology. The research was performed by M. Stöhr and M. Sadhukhan. All authors contributed to the interpretation of the results. The original draft of this manuscript was written by M. Stöhr, M. Sadhukhan, and Y.S.A., the final manuscript by M. Stöhr with contributions from all authors. A.T. supervised the project.

## Competing interests
The authors declare no competing interests.
