## [Peer Review File · Nature Communications]

REVIEWER COMMENTS

Reviewer #1 (Remarks to the Author):

I am pleased to submit my review on "Coulomb Interactions between Dipolar Quantum Fluctuations in van der Waals Bound Molecules and Materials" by Stohr et al for consideration in Nature Communications. While I think the work will be very interesting to physical chemists and chemical physicists, I do not consider it to be suitable for the broad audience of Nat Comm. I instead recommend submission to a more technical journal, after expanding the technical discussion.

Firstly, I should note that I think the presented work is very interesting. The authors have taken QHO perturbation theory to the next level (their CS correction) and shown that doing so can be important in systems of interest. The manuscript is very clearly explained, for an inexpert reader; although it does require, in my opinion, technical discussion to be expanded for completeness. If its results were substantially more compelling I might recommend that Nat Comm publish it.

Unfortunately, I simply do not find the results sufficiently compelling. Certainly they are interesting. But I feel that they do not stand up well to critical scrutiny, so would run the risk of painting a false picture for inexpert readers who do not have the "bigger picture" of QHO dispersion theory to draw from while interpreting the results. I am fairly certain that some of the figures could be equally well reproduced by other tweaks to the underlying MBD model, which contains a large number of assumptions.

In terms of theory, I think it is based on an unjustified assumption. Real dispersion interactions come from localised fluctuations of weakly bound electrons, almost the opposite of QHOs. The reason QHOs work so well is that they can be tuned to reproduce the leading order behaviour of those electrons. There is no especially good reason to think, however, that the same is true for next-leading order. I don't feel the manuscript addresses this point at all.

In terms of results, they start well with good agreement presented in Figure 2 between the CS corrected MBD and benchmark QMC reference values for two systems. But the shown improvements could be matched by any correction that simply added ~ 7 kJ/mol to both systems, e.g. by changing the polarisability model or changing the damping function. What would be needed to make this point more believable is a demonstration that a similar system with smaller CS terms was also improved, which one must wait until Figure 3 to see. [note, I am assuming that the target is the solid blue line]

However, in Figure 3 the improvement is worse. The CS correction in R2 and R3 is much smaller than what is required to match QMC - I suspect that other reasonable tweaks to QHO models (e.g. to parameters or spatial extent) would induce similar changes. R3 and R4 have similar measures of distortion and proximity, but R3 is out by ~ 8 kcal/mol and corrected by ~ 2 kcal/mol, whereas R4 is out by 4 kcal/mol and corrected by the same. Why is this? The authors offer an explanation by invoking global geometry considerations in the quantum correlations, which would be fine (it is probably true) if the numbers were better. Here it just feels like they had some bad cases and invoked "quantum" to explain them.

Figure 4 is interesting, but seems similar to the cases presented in [32]. It is also not supported by QMC calculations, meaning it is essentially an argument about what the model can do. These plots really need to include the PBE0 energies too (suitably scaled) to convince the reader that quadrupoles induced by charge asymmetries at the PBE0 level are not of similar magnitude.

None of these issues would be so problematic if the method was truly *ab initio*, i.e. it represented a convergible step toward an exact solution. But it is not. It is an approximate correction (CS) to an approximate model (MBD, QHO) sitting atop another approximation (PBE0). It is extremely nice in its context. But its context is relatively narrow and its results of general appeal are not sufficiently compelling to make up for its limitations.

In summary, the manuscript is well written and the work presented is interesting. I have no major issues with it other than a lack of technical detail. But I think it would be premature to present these results in Nat Comm.

Reviewer #2 (Remarks to the Author):

In this contribution, Stohr et al. proposed to study the coulomb Interactions arising between dipolar Quantum fluctuations in van der Waals bound Molecules and materials. The paper is extremely interesting and deserves publication in Nature Communications. I have only a few comments that could help the authors (the paper could be published as it).

i) As a quantum chemist, I think that the "Coulomb Singles" is definitively not the best way to "name" the new object that is here described. As the authors note, "the present Singles term is to be distinguished from those in post-Hartree-Fock methods or the Random Phase Approximation, where the zeroth order theory correspond to a mean-field, uncorrelated state." Ok, I totally agree but these different meanings will have to cohabit leading to potential confusion. Maybe something could be done here.

ii) concerning the higher order multipoles, it would be worth to discuss a little but further the "where to stop" question. The authors send the reader to their previous works but as the paper is really compact and include lots of information, it would be good to organize somewhere a more self-contained discussion.

iii) How fast it is computationally? Could it be used in force fields?

Overall, this paper is a major contribution to the field.

We would like to thank the reviewers for their work and very helpful comments to our manuscript. The revised manuscript most prominently features an extended discussion of the coupled oscillator model for electronic response and the presented formalism in a greater context. As a general remark, we would like to note that the term “Coulomb Singles (CS)” has been changed to “Dipole-Correlated Coulomb Singles (DCS)” in order to reflect the described physics more clearly. You find a detailed point-by-point reply to the individual comments below.

Reviewer #1:

I am pleased to submit my review on “Coulomb Interactions between Dipolar Quantum Fluctuations in van der Waals Bound Molecules and Materials” by Stöhr *et al.* for consideration in Nature Communications. While I think the work will be very interesting to physical chemists and chemical physicists, I do not consider it to be suitable for the broad audience of Nat Comm. I instead recommend submission to a more technical journal, after expanding the technical discussion.

Firstly, I should note that I think the presented work is very interesting. The authors have taken QHO perturbation theory to the next level (their CS correction) and shown that doing so can be important in systems of interest. The manuscript is very clearly explained, for an inexpert reader; although it does require, in my opinion, technical discussion to be expanded for completeness. If its results were substantially more compelling I might recommend that Nat Comm publish it.

Reply: We have extended the discussion of conventional QHO response/dispersion theory as well as connections and differences to the presented DCS formalism to the revised manuscript. This includes further explanation of the terms present in the DCS energy expression, a distinction from conventional multipolar expansions of van der Waals interactions, the justification for limiting the presented formalism to first-order contributions, as well as a brief explanation of the broader context and limit of the employed perturbation series.

Unfortunately, I simply do not find the results sufficiently compelling. Certainly they are interesting. But I feel that they do not stand up well to critical scrutiny, so would run the risk of painting a false picture for inexpert readers who do not have the “bigger picture” of QHO dispersion theory to draw from while interpreting the results. I am fairly certain that some of the figures could be equally well reproduced by other tweaks to the underlying MBD model, which contains a large number of assumptions.

Reply: We would like to remark that the only “tweaks” of the underlying MBD model are the definition of effective atomic polarizabilities *via* Hirshfeld partitioning (which can be expected to perform well in non-metallic, organic systems without strongly localized charges as considered in this work) and the damping function. As discussed in further detail below, the observed results describe so-far neglected physics of van der Waals dispersion, which cannot be captured or described effectively by trivial modifications to the damping function. As detailed in the literature [*e.g.*, *Chem. Soc. Rev.* **48**, 4118 (2019) or *Chem. Rev.* **117**, 4714 (2017)], the remaining formalism is fully in line with the random phase approximated (long-range) correlation energy for the coupled oscillator model as given by the adiabatic-connection fluctuation-dissipation theorem. The suitability of the QHO response model for molecular systems is discussed below.

In terms of theory, I think it is based on an unjustified assumption. Real dispersion interactions come from localised fluctuations of weakly bound electrons, almost the opposite of QHOs. The reason QHOs work so well is that they can be tuned to reproduce the leading order behaviour of those electrons. There is no especially good reason to think, however, that the same is true for next-leading order. I don't feel the manuscript addresses this point at all.

Reply: The QHO model, in fact, describes exactly such localized fluctuations of weakly bound (valence) electrons. In the case of non-metallic systems as considered here, mapping these fluctuations onto atomic polarizability centers can be considered a valid approach. Note that the finite, capped armchair carbon nanotubes studied in the last part of the manuscript still show a sufficient HOMO-LUMO gap of ~ 0.8 eV (PBE0). As the reviewer points out, the QHO model performs very well in reproducing the leading-order behavior of real atoms. Beyond this leading-order behavior, we can further corroborate that the coupled QHO/QDO model for electronic response

- provides highly accurate molecular polarizabilities and C_6 -interaction coefficients for a wide range of molecules and materials at a mean absolute relative error of less than 10 % for small molecules, chains of hydrogen molecules and silicon clusters as well as a quantitatively correct scaling in semiconductor solids [see *J. Phys.: Condens. Matter* **26**, 213202 (2014), *J. Chem. Phys.* **140**, 18A508 (2014), *Chem. Rev.* **117**, 4714 (2017)]
- allows to accurately describe both, polarization and dispersion, in realistic systems [e.g., *Phys. Rev. B* **87**, 144103 (2013) and *J. Chem. Theory Comput.* **9**, 5430 (2013)]
- reproduces the relation between electronic polarizability and geometry in van der Waals-bound dimers as obtained for real atoms within an error of $< 4\%$ [*Phys. Rev. Lett.* **121**, 183401 (2018)]
- provides a reliable description of excess electrons in matter [*J. Phys. Chem. B* **117**, 4365 (2013)]
- quantitatively captures electron redistribution due to long-range correlation in comparison to fully self-consistent dispersion-polarized DFT calculations [*Nat. Commun.* **8**, 14052 (2017)]
- can be seamlessly applied to molecules as well as solids [*Phys. Rev. Lett.* **124**, 146401 (2020)]

On a more fundamental level, we would like to point out that an infinite set of dipole-coupled QHOs can essentially describe any response allowed by quantum field theory and can thus model the response of arbitrary molecules or materials. When employing beyond-dipolar coupling, a discrete set of (coarse-grained) oscillators then effectively provides an equivalent framework. We have added a brief introduction to the QHO model and the above background information in condensed form to the revised manuscript.

In terms of results, they start well with good agreement presented in Figure 2 between the CS corrected MBD and benchmark QMC reference values for two systems. But the shown improvements could be matched by any correction that simply added ~ 7 kJ/mol to both systems, e.g. by changing the polarizability model or changing the damping function. What would be needed to make this point more believable is a demonstration that a similar system with smaller CS terms was also improved, which one must wait until Figure 3 to see. [note, I am assuming that the target is the solid blue line]

However, in Figure 3 the improvement is worse. The CS correction in R2 and R3 is much smaller than what is required to match QMC - I suspect that other reasonable tweaks to QHO models (e.g. to parameters or spatial extend) would induce similar changes. R3 and R4 have similar measures of distortion and proximity, but R3 is out by 8 kcal/mol and corrected by 2 kcal/mol, whereas R4 is out by 4 kcal/mol and corrected by the same. Why is this? The authors offer an explanation by invoking global geometry considerations in the quantum correlations, which would be fine (it is probably true) if the numbers were better. Here it just feels like they had some bad cases and invoked “quantum” to explain them.

Reply: As detailed in the text, the DCS contributions reported in Fig. 3 are 6.4 and 4.9 kJ/mol. So, a “correction that simply added ~ 7 kJ/mol to both systems” would not correctly reproduce this behavior. Based on the results given for the S66 data set and the different ring systems shown in Fig. 4, it should also be apparent that the DCS contribution is highly system-dependent and no constant, general reduction of the interaction energy as trivial changes in the damping function or polarizability model would introduce. The results reported for Xe₂ in CNTs, then, give a final confirmation that a simple modification of the damping function cannot capture the complex changes introduced by extending the treatment of van der Waals forces *via* DCS. Admittedly, PBE0+MBD+DCS does not reach the QMC reference data for the complex “R3”. This is the only case, however, and DCS still modify the interaction energy in the right direction [given the stochastic nature of (D)QMC, any value within the blue box can be considered the correct result]. So, in all cases but one, inclusion of DCS allows to reach QMC level of accuracy at a fraction of the computational costs. Note that the lack of correlation between simple geometric descriptors and the DCS contribution that the reviewer describes would still remain when disregarding this one case, where no quantitative agreements is reached. As such, we argue that the DCS contribution cannot be described by simple geometrical considerations and thus trivial classical corrections or modifications of the damping function.

Figure 4 is interesting, but seems similar to the cases presented in [32]. It is also not supported by QMC calculations, meaning it is essentially an argument about what the model can do. These plots really need to include the PBE0 energies too (suitably scaled) to convince the reader than quadrupoles induced by charge asymmetries at the PBE0 level are not of similar magnitude.

Reply: Ref. 32 reports the scaling behavior of a beyond-dipolar correction over dipole-coupled oscillators for the cases of idealized low-dimensional harmonic oscillators. The present results within the DCS formalism, on the other side, are reported for the realistic system of an actual Xe dimer inside carbon nanotubes. As described in the manuscript, the present work advances the previous, academic study of model systems [32] to be applicable in atomistic modeling as well as to confirm and highlight the role of the so-far neglected physics in the van der Waals interaction of realistic systems. Concerning PBE(0) energies for Xe₂ in CNTs, we firstly would like to point out that the DCS contribution is additive in this regard. So, the differences in the total long-range interaction (with vs without DCS) are regardless of the PBE0 energetics. In the case of the PBE calculations, that were performed for this work, the mean-field DFT contribution is negligible beyond *inter*-Xe distances of 6 Å and only introduces repulsive contributions towards shorter distances. As a result, the qualitative differences reported for the total van

der Waals interaction in Fig. 4 are not altered by the addition of PBE-DFT energies. One interesting example when considering the full DFT-inclusive interaction energy is that for the Xe dimer in the (5,5)-CNT, the DCS contribution cancels out the local minimum around 5.5 Å, where PBE+MBD predicts an equilibrium state. This important discussion has been added to the revised manuscript. In addition, we added the results for PBE, PBE+MBD & PBE+MBD+DCS to the supplemental information.

None of these issues would be so problematic if the method was truly ab initio, i.e. it represented a convergable step toward an exact solution. But it is not. It is an approximate correction (CS) to an approximate model (MBD, QHO) sitting atop another approximation (PBE0). It is extremely nice in its context. But its context is relatively narrow and its results of general appeal are not sufficiently compelling to make up for its limitations.

Reply: We agree that the presented formalism might not strictly converge to the exact solution on a theoretical basis. It does, however, represent a decisive step towards a more complete treatment of van der Waals forces, while still being applicable to complex systems. In this regard, it offers a significant advantage over current state-of-the-art approaches with comparable accuracy and completeness, which do not allow for such applications due to their much higher computational costs. We would like to emphasize that the DCS formalism does not represent a mere correction based on physical reasoning, but has a clear and well-defined limit given by fully Coulomb-coupled quantum (Drude) oscillators. This model has been widely used and shown to provide a highly sophisticated treatment of van der Waals dispersion [see, e.g., *Phys. Rev. B* **87**, 144103 (2013)]. These points have been clarified in the extended discussion of the QHO dispersion model in the context of the presented methodology.

In summary, the manuscript is well written and the work presented is interesting. I have no major issues with it other than a lack of technical detail. But I think it would be premature to present these results in Nat Comm.

Reply: Considering the above explanations and clarifications, we are convinced (as is Reviewer #2) that the so-far neglected aspect of van der Waals dispersion reported in this work and the well-defined DCS formalism, in fact, do provide an important step forward in describing and understanding van der Waals forces in complex systems, which is not accessible with current state-of-the-art approaches. As such, we also believe the present work appeals to a broad audience in physical, chemical, and biomolecular sciences.

Reviewer #2:

In this contribution, Stöhr *et al.* proposed to study the coulomb Interactions arising between dipolar Quantum fluctuations in van der Waals bound Molecules and materials. The paper is extremely interesting and deserves publication in Nature Communications. I have only a few comments that could help the authors (the paper could be published as it).

- i) As a quantum chemist, I think that the “Coulomb Singles” is definitively not the best way to “name” the new object that is here described. As the authors note, “the present Singles term is to be distinguished from those in post-Hartree-Fock methods or the Random Phase Approximation, where the zeroth order theory correspond to a mean-field, uncorrelated state.” Ok, I totally agree but these different meanings will have to cohabit leading to potential confusion. Maybe something could be done here.

Reply: We admit that the usage of the term “Singles” can potentially lead to confusion. After reconsidering, we have changed the name to “Dipole-Correlated Coulomb Singles (DCS)”, which hopefully attenuates potential confusion by clarifying that the present “Singles” contributions are evaluated on a (dipole-)correlated state. The term “Singles” is further motivated by a likely resemblance of the DCS contribution and singles amplitudes in configuration interaction theory. For now, we would prefer to not include this in the manuscript, however, as a clear connection between the two is still not fully established and subject to ongoing work.

- ii) concerning the higher order multipoles, it would be worth to discuss a little but further the “where to stop” question. The authors send the reader to their previous works but as the paper is really compact and include lots of information, it would be good to organize somewhere a more self-contained discussion.

Reply: The discussion of QHO response/dispersion theory as well as the similarities and differences of the presented formalism to conventional interatomic approaches has been largely extended in the revised manuscript. This should give helpful background information for the non-expert reader, highlight the solid theoretical foundation of the DCS formalism, and also justify the present limitation to the first-order term.

- iii) How fast it is computationally? Could it be used in force fields?

Reply: Straightforward computation of DCS without approximations as presented in this manuscript scales with the fifth power of the number of atoms. The computational prefactor, on the other side, is very small (no time-consuming integrations at run time, no basis set expansion and independent of the number of electrons per atom). As a result, the DCS formalism introduces negligible computational costs compared to semi-local or hybrid DFT up to systems with several hundreds of atoms. Further developments towards cubic scaling and an analysis of when the additional computational costs may become a limiting step are in progress. Nonetheless, the presented formalism is computationally much less demanding than all other electronic structure methods that are able to capture the DCS contributions and only adds a minimal overhead to van der Waals-inclusive DFT calculations. DCS solely rely on the dipolar-coupled MBD wave function, which in turn can be obtained based on the knowledge of atomic

polarizabilities. When combined with a reliable scheme to obtain effective atomic response properties [see *J. Chem. Phys.* **141**, 1 (2014), for example], DCS contributions can, in principle, also be included in force fields. While the remaining computational costs of the overall scheme can considerably limit typical applications such as molecular dynamics simulations, the formalism can be used for the energy reweighting of structural ensembles. In order to clarify this point for a broader audience in the molecular modeling community, this discussion has been added to the manuscript.

Overall, this paper is a major contribution to the field.

REVIEWERS' COMMENTS

Reviewer #1 (Remarks to the Author):

I am pleased to submit my review on the revised manuscript "Coulomb Interactions between Dipolar Quantum Fluctuations in van der Waals Bound Molecules and Materials" by Stohr et al for consideration in Nature Communications. I feel the revised manuscript is much more honest about the reported advances and their limitations and could be published in Nature Communications, subject to minor revisions.

The one change I would like to see the author's make is to introduce some discussion about how the work presented in the manuscript relates to self-consistent RPA in the discussion about RPA, to address the question of whether sc-RPA captures these effects. I suspect not but feel it would be the final missing piece to connect the coarse-grained model with response theory.

Reviewer #1:

I am pleased to submit my review on the revised manuscript “Coulomb Interactions between Dipolar Quantum Fluctuations in van der Waals Bound Molecules and Materials” by Stöhr *et al.* for consideration in Nature Communications. I feel the revised manuscript is much more honest about the reported advances and their limitations and could be published in Nature Communications, subject to minor revisions.

The one change I would like to see the author’s make is to introduce some discussion about how the work presented in the manuscript relates to self-consistent RPA in the discussion about RPA, to address the question of whether sc-RPA captures these effects. I suspect not but feel it would be the final missing piece to connect the coarse-grained model with response theory.

Reply: We would like to thank the reviewer for raising this excellent point missing from the previous discussion. An accurate, fully self-consistent treatment of electron correlation within the RPA should indeed allow to capture the contributions of Dipole-Correlated Coulomb Singles (DCS). It has been shown, however, that current implementations of the self-consistent RPA approach, like the self-consistent *GW* method [F. Caruso, P. Rinke, X. Ren, M. Scheffler and A. Rubio, *Phys. Rev. B* **86**, 081102 (2012)], do not yet capture the charge density polarization induced by long-range correlation forces on a quantitatively accurate level (see N. Ferri, R.A. DiStasio Jr., A. Ambrosetti, R. Car and A. Tkatchenko, *Phys. Rev. Lett.* **114**, 176802 (2015), for example). We are confident that future work and further developments of this complementary approach will enable to accurately capture the effects described in this work as well as to identify further connections between the coarse-grained model and the more established wave function- or response theory-based formalisms.

The above discussion on sc-RPA as well as its current limitations in the context of DCS have been added to the revised manuscript.